# The Association between Vitamin D and Zinc Status and the Progression of Clinical Symptoms among Outpatients Infected with SARS-CoV-2 and Potentially Non-Infected Participants: A Cross-Sectional Study

**DOI:** 10.3390/nu13103368

**Published:** 2021-09-25

**Authors:** Sahar Golabi, Maryam Adelipour, Sara Mobarak, Maghsud Piri, Maryam Seyedtabib, Reza Bagheri, Katsuhiko Suzuki, Damoon Ashtary-Larky, Fatemeh Maghsoudi, Mahshid Naghashpour

**Affiliations:** 1Department of Medical Physiology, School of Medicine, Abadan University of Medical Sciences, Abadan 6313833177, Iran; s.golabi@abadanums.ac.ir; 2Department of Clinical Biochemistry, School of Medicine, Ahvaz Jundishapur University of Medical Sciences, Ahvaz 6135715794, Iran; adelipour-m@ajums.ac.ir; 3Department of Infectious Diseases, School of Medicine, Abadan University of Medical Sciences, Abadan 6313833177, Iran; s.mobarak@abadanums.ac.ir; 4Vice Chancellor for Health, Abadan University of Medical Sciences, Abadan 6313833177, Iran; maghsudpiri@gmail.com; 5Department of Biostatistics and Epidemiology, School of Public Health, Ahvaz Jundishapur University of Medical Sciences, Ahvaz 6135715794, Iran; m.stabib3@gmail.com; 6Department of Exercise Physiology, University of Isfahan, Isfahan 8174673441, Iran; will.fivb@yahoo.com; 7Faculty of Sport Sciences, Waseda University, 2-579-15 Mikajima, Tokorozawa 359-1192, Japan; katsu.suzu@waseda.jp; 8Nutrition and Metabolic Diseases Research Center, Ahvaz Jundishapur University of Medical Sciences, Ahvaz 6135715794, Iran; damoon_ashtary@yahoo.com; 9Department of Public Health, School of Health, Abadan University of Medical Sciences, Abadan 6313833177, Iran; fatemehmagh627@gmail.com; 10Department of Nutrition, School of Medicine, Abadan University of Medical Sciences, Abadan 6313833177, Iran

**Keywords:** clinical symptoms, vitamin D status, zinc status, sunlight exposure, COVID-19

## Abstract

Vitamin D and zinc are important components of nutritional immunity. This study compared the serum concentrations of 25-hydroxyvitamin D (25(OH)D) and zinc in COVID-19 outpatients with those of potentially non-infected participants. The association of clinical symptoms with vitamin D and zinc status was also examined. A checklist and laboratory examination were applied to collect data in a cross-sectional study conducted on 53 infected outpatients with COVID-19 and 53 potentially non-infected participants. Serum concentration of 25(OH)D were not significantly lower in patients with moderate illness (19 ± 12 ng/mL) than patients with asymptomatic or mild illness (29 ± 18 ng/mL), with a trend noted for a lower serum concentration of 25(OH)D in moderate than asymptomatic or mild illness patients (*p* = 0.054). Infected patients (101 ± 18 µg/dL) showed a lower serum concentration of zinc than potentially non-infected participants (114 ± 13 µg/dL) (*p* = 0.01). Patients with normal (odds ratio (OR), 0.19; *p* ≤ 0.001) and insufficient (OR, 0.3; *p* = 0.007) vitamin D status at the second to seventh days of disease had decreased OR of general symptoms compared to patients with vitamin D deficiency. This study revealed the importance of 25(OH)D measurement to predict the progression of general and pulmonary symptoms and showed that infected patients had significantly lower zinc concentrations than potentially non-infected participants.

## 1. Introduction

Novel coronavirus disease 2019 (COVID-19), caused by severe acute respiratory syndrome coronavirus 2 (SARS-CoV-2), is a worldwide pandemic that originally emerged in Wuhan, China [1]. As of 3 April 2020, Iran has been among the countries with the highest burden of the COVID-19 outbreak [2]. COVID-19 is characterized by the symptoms of viral pneumonia, such as fever, fatigue, dry cough, and lymphopenia. Patients have reported comorbidities such as diabetes, cardiovascular disease, liver disease, kidney disease, and malignant tumors [3]. This disease also affects physical activity, sedentary action, and psychological emotion [4].

While therapeutic options are still under investigation, and some vaccines have been approved, cost-effective ways to reduce the probability of or even prevent infection and the shift from mild symptoms to more serious detrimental disease are highly worthwhile [5].

An appropriate diet and good nutritional status are essential for an optimal immune response to prevent infections. On the other hand, a poor diet and deficiency of these nutrients will increase the disease burden. Evidence proposes that nutrients are involved in the development of COVID-19 [6]. 

Vitamin D3 is a pre-pro-hormone that begins its biosynthesis pathway with the solar UVB irradiation of 7-dehydrocholesterol on bare skin exposed to strong sunlight and exhibits multifaceted effects beyond calcium and bone metabolism. Vitamin D is essential to balance immune responses [7]. Since vitamin D receptors are expressed on immune cells (B, T, and antigen-presenting cells), which can synthesize the active metabolite of vitamin D, this vitamin can act in an autocrine manner in a local immunological environment. Iran is a country with a high prevalence of vitamin D deficiency among various age groups, with the more apparent prevalence of this deficiency in Tehran, the capital of Iran. According to a systematic review and meta-analysis, the prevalence rate of vitamin D deficiency among the Iranian population is wide-ranging from 2.5% to 98% in various studies and regions [8]. Vitamin D deficiency is associated with increased autoimmunity and increased susceptibility to infection [9]. 

Epidemiological evidence suggests a significant association between vitamin D deficiency and an increased incidence of several infectious diseases, viral respiratory tract infections [10], and influenza [11]. A recent epidemiologic study reported a strong significant relationship between the serum concentration of vitamin D and the number of deaths per million people from COVID-19 across 20 European countries [12]. Previous findings have shown that individuals with vitamin D deficiency have a higher risk of contracting a severe COVID-19 disease [13].

It is well-known that zinc is a critical mineral in many biological processes due to its functions as a cofactor, signaling molecule, and structural element [14]. Furthermore, zinc has an important role in the regulation of the immune system by regulating the proliferation, differentiation, maturation, and functioning of leukocytes and lymphocytes [15]. Zinc also plays a signaling role in the modulation of inflammatory responses [16]. It is also a component of nutritional immunity [17]. Previously published data demonstrate that zinc status is associated with the prevalence of respiratory tract infections in children and adults [18,19]. It is also thought that zinc has the potential to support COVID-19 therapy due to its immunomodulatory roles and direct antiviral effects [20].

Moreover, adequate dietary intake of zinc and vitamin D is essential for suitable immunocompetence and resistance to viral infections [21]. In addition, an ecological study demonstrates that intake levels of vitamin D are inversely accompanied by higher COVID-19 incidence and/or mortality, especially in populations that are genetically predisposed to low micronutrient status [22]. Moreover, it is suggested that nutrition intervention acquiring an adequate status of some vitamins and minerals including vitamin D and zinc might protect against COVID-19 and alleviate the course of the disease [21,22]. However, dietary recommendations alone are not enough to ensure the adequacy of these nutrients [21]. As a result, nutritional data assessing nutrients are essential for immune system function [22]. 

So far, data on the association between vitamin D and zinc status and the progression of symptoms during the clinical course among COVID-19 outpatients are limited. The high prevalence of vitamin D and zinc deficiency in the elderly, smokers, patients with chronic diseases, and obese individuals suggests that vitamin D and zinc play a role as therapeutic agents against COVID-19. Here, we evaluated the role of the nutritional status of vitamin D and zinc in the perspective of COVID-19 and the progression of symptoms during the clinical course of the disease. Therefore, we compared the demographics, baseline comorbidities, and serum concentrations of vitamin D and zinc at the second to seventh days of disease between infected outpatients with COVID-19 and potentially non-infected participants from an academic health care setting in southwestern Iran. The association between serum concentrations of vitamin D and zinc at the second to seventh days of disease and the progression of symptoms during the clinical course of the disease was also determined. 

## 2. Materials and Methods

### 2.1. Study Design 

To examine the potential association between vitamin D and zinc status and the disease progression of COVID-19 among the clients of a health care setting, we designed a health service center-based cross-sectional and descriptive–analytical study aimed to compare infected outpatients with COVID-19 and potentially non-infected participants in terms of demographics, baseline comorbidities, and serum concentrations of vitamin D and zinc at the second to seventh days of disease. In addition, patients who tested positive for COVID-19 by reverse transcription–polymerase chain reaction (RT-PCR) were followed up from day 1 to day 28 after the onset of symptoms to evaluate the effect of vitamin D and zinc status at the second to seventh days of disease on the symptom progression during the clinical course of COVID-19. This research was approved by the Ethics Committee of Abadan University of Medical Sciences (Ethics code: IR.ABADANUMS.REC.1399.073). The criterion for entering the infected and potentially non-infected participants was a positive or negative RT-PCR result. Infected patients: patients with a laboratory-confirmed COVID-19 diagnosis based on the RT-PCR test. Potentially non-infected participants: individuals whose COVID-19 had not been confirmed based on the RT-PCR test with no history of positive RT-PCR test during the COVID-19 pandemic or recovering COVID recently, no clinical signs associated with COVID-19, including high fever, and high-risk occupations, including medical staff. 

### 2.2. Setting

Sixteen-hour COVID-19 health service centers operate under the supervision of Abadan University of Medical Sciences. These centers were activated following the outbreak of the COVID-19 pandemic and work on an outpatient basis due to the need to provide health services for the citizens of Abadan (located in southwestern Iran). Outpatients with COVID-19 and potentially non-infected participants referred to centers from 6 June 2020, to 12 August 2020, were recruited in the study. 

### 2.3. Study Population and Sample

The population of this study comprised clients referred to the 16-hour outpatient centers mentioned in the previous section. All participants provided written informed consent during recruitment for study participation and repeat contact. All clinical investigations were conducted according to the ethical standards of the World Medical Association’s Declaration of Helsinki. Infected patients were at the second to seventh days of COVID-19 disease.

Age- and sex-matched potentially non-infected participants with negative RT-PCR test results were recruited from the same 16-hour health service center by telephone and underwent screening by a study team member. 

All infected patients and potentially non-infected participants underwent respiratory sampling, including nasal and pharyngeal swabs, bronchoalveolar lavage fluid, sputum, or bronchial aspirates, in one of the 16-hour outpatient centers to evaluate COVID-19. An RT-PCR kit (COVITECH, Tehran, Iran) was used to qualitatively detect the presence or absence of *SARS*-*CoV*-*2* infection, which is currently used in the Iranian health centers to diagnose COVID-19 disease. Cut-off Ct value < 36 was considered as a positive result. 

We used an open-source calculator to calculate the minimum sample size required based on the probability of a type I error of alpha = 0.5 and type II error of beta = 0.2 (power = 80%). According to this calculation, at least 53 cases and 53 controls were needed. Individuals with a clear RT-PCR result (either positive or negative) meeting the essential criteria to enter the study were selected by a simple sampling method, so that every client had an equal probability of admission and inclusion in the study. We used the demographic factors of age and sex as factors to ensure that our potentially non-infected participants were matched with our infected patients. As such, a potentially non-infected participant with a specific age and sex was included in the study for each infected patient of the same age and sex.

### 2.4. Inclusion and Exclusion Criteria

Participants ≥11 years of age of both sexes were included in the study. Moreover, to be included, participants needed to have a clear RT-PCR result (positive or negative) and be willing to participate in the study. They also needed to have the ability to understand the relevant information and complete the informed consent form. Pregnant and lactating women, participants with uncertain RT-PCR test results, and patients with sickle cell anemia or thalassemia were excluded [23]. 

### 2.5. Variables 

The variable presented a positive result for the specific test for COVID-19 detection. Moreover, to identify the stage of COVID disease, infected patients were categorized according to disease severity and prognosis using Center for Disease Control and Prevention (CDC) criteria, which include the following. (1) *Asymptomatic or presymptomatic disease*: individuals who presented positive results for the RT-PCR test but showed no symptoms of COVID-19. (2) *Mild illness*: individuals who had any of the symptoms of COVID-19 (e.g., fever, headache, cough, sore throat, muscle pain, malaise, vomiting, nausea, diarrhea, and smell and taste disorders) but did not have dyspnea, shortness of breath, or abnormal chest imaging. (3) *Moderate illness*: individuals who indicated evidence of lower respiratory disease during clinical assessment or imaging and oxygen saturation (SpO_2_) of ≥94% in room air at sea level. (4) *Severe illness*: individuals who had an SpO_2_ of <94% in room air at sea level, a pressure of oxygen to fraction of inspired oxygen (PaO_2_/FiO_2_) of <300 mm Hg, a respiratory frequency of >30 breaths/min, or lung infiltrates at >50%. (5) *Critical illness*: individuals with septic shock, respiratory failure, and/or multiple organ dysfunction [24].

In the present study, no infected patients with severe or critical diseases were found among the participants. Furthermore, *asymptomatic* and *mild* categories were defined as “mild and no sign” in the data analysis.

Primary outcomes were based on clinical and laboratory examinations, as well as exposure to sunlight; secondary outcomes were related to clinical symptoms. Additionally, demographic evidence (age, sex, marital status, education level, and smoking habits), comorbidities, body mass index (BMI), and taking nutritional supplements were potential confounders. 

### 2.6. Data Sources and Measurements

After the RT-PCR test results were determined, a checklist was given to all infected and potentially non-infected participants so that they could provide information on demographic and anthropometric characteristics, signs and symptoms, current smoking status, and any comorbidities or other conditions that have been linked to the disease (e.g., cardiovascular disease, diabetes mellitus, chronic obstructive pulmonary disease and other lung diseases, cancers, chronic kidney disease, obesity, taking nutritional supplements, and smoking) [25]. In addition, sunlight exposure was quantified through a questionnaire as a proxy measure for vitamin D status [26].

Clinical examinations including respiratory rate (RR), pulse rate (PR), and SpO_2_ levels were measured by a pulse oximeter at the time of admission on day 1 (second to seventh days of disease) in the health service center. 

Laboratory examination including serum concentrations of a total of 25-hydroxyvitamin D (25(OH)D) and zinc was conducted on the admission day. After informed consent had been obtained, around 5 mL of blood was collected following 8 h of fasting. Biochemical analysis was performed on the serum sample after separation, and the serum concentrations of zinc were measured with a fully automated analyzer (Miura, ISE Co., Italy) using a kit for the quantitative determination of the zinc according to the manufacturer’s protocol (PaadCo Co., Iran) following a direct colorimetric method. The reference values used for serum concentration of zinc were 68–107 μg/dL. To assess the whole-body vitamin D status of the participants, serum concentrations of a total of 25(OH)D were measured retrospectively in serum samples collected in gel tubes at the time of admission. Serum concentrations of 25(OH)D were quantified using a commercially available immunoassay (Vitamin D 96 ELISA Kit. Ideal, Ideal Tashkhis Ateieh, Tehran, Iran). The mean inter-assay coefficients of variation (CVs) for the 25(OH)D and zinc concentrations were 8.3% and 5.9%, respectively. Intra-assay CVs were not conducted for 25(OH)D and zinc measurements.

However, all experiments were performed in a clinical laboratory having a quality control certificate from Iran Health Reference Laboratory. The procedure of 25(OH)D measurement in the serum has been illustrated in Appendix A.

We used further stratification for the serum concentrations of 25(OH)D and categorized infected patients and potentially non-infected participants in terms of serum concentrations of 25(OH)D to normal, insufficient, and deficient vitamin D status, so that the cut-off point of 25(OH)D 12–20 ng/mL (30–50 nM) was defined as vitamin D insufficiency, and <12 ng/mL (equivalent to <30 nM) as vitamin D deficiency. Additionally, a cut-off point of >20 ng/mL (>50 nM) was defined as normal. This categorization was according to the criteria of the Institute of Medicine (US) Committee to Review Dietary Reference Intakes for Vitamin D and Calcium Dietary Reference Intakes for Calcium and Vitamin D [27]. This categorization was used to compare the vitamin D status between infected patients and potentially non-infected participants.

The data of clinical symptoms were collected from both asymptomatic and symptomatic COVID-19-infected patients to evaluate the disease progression by recording self-reported health information weekly. The recorded information included the symptoms and pre-existing medical conditions obtained on day 1 at the sampling site and then on days 7, 14, 21, and 28 of the first symptoms observed by telephone contact. It was assumed that individuals with negative RT-PCR results, no clinical signs or symptoms of COVID-19, and no high-risk occupations (e.g., medical staff, taxi drivers) were not infected. Commonly presented clinical symptoms of COVID-19 fell into four categories: (1) general (fatigue, fever, night sweats, asthenia, flushing, chills, hypothermia, runny nose, sore throat), (2) pulmonary (chest pain, shortness of breath, dyspnea, cough), (3) gastrointestinal (anorexia, abdominal cramps, diarrhea, vomiting, nausea, constipation, bloating), and (4) neurologic (headache, muscle pain, joint pain, ear pain, new smell and taste disorders such as anosmia and dysgeusia) [28]. 

### 2.7. Statistical Analysis 

We matched the data of the infected patients with those of the potentially non-infected individuals of the same sex and age. Testing of data for normal distribution was carried out using the Kolmogorov–Smirnov test. Characteristics of the infected patients and potentially non-infected participants were compared using the χ² test for discrete variables and the independent sample *t*-test for continuous variables. A generalized estimating equation (GEE) regression model with a logistic link function and an exchangeable correlation structure for each individual was employed to assess the odds ratio (OR) and 95% confidence interval (95% CI) of the disease symptoms on days 1, 7, 14, 21, and 28 after the onset of the first symptoms. GEE model was restricted to patients infected with SARS-CoV-2. The model was adjusted for potential confounding variables including age, sex, marital status, education levels, and BMI. 

All descriptive analyses and the GEE modeling were performed using IBM SPSS Statistics (version 26). In all tests, *p* < 0.05 was considered to indicate statistical significance. 

## 3. Results

### 3.1. Participants’ Characteristics 

As illustrated in Figure 1, a total of 1181 potentially eligible clients were admitted to the health centers. Among them, 1169 clients with a confirmed RT-PCR test result (691 clients with positive and 478 clients with negative RT-PCR test results) visited the health service center from 6 June 2020 to 12 August 2020 and their eligibility was confirmed. Following the simple randomization by telephone call and matching in terms of age and sex, a total of 108 individuals (54 infected patients and 54 potentially non-infected participants) contributed to the study and blood sampling. One infected patient was excluded from the study following the diagnosis of pregnancy. Additionally, one potentially non-infected participant was excluded due to unwillingness to continue the study. Ultimately, 53 infected patients (male = 68%; mean age = 41 years) and 53 age- and sex-matched potentially non-infected participants (male = 72%; mean age = 40 years) completed the follow-up and analysis. 

The infected patients’ and potentially non-infected participants’ characteristics in the study are given in Table 1. There was no significant difference in mean age among infected patients and potentially non-infected participants. Participants were predominantly male and had no significant differences in terms of their marital status, education level, cigarette smoking status, comorbidities, and BMI. Additionally, respiratory rate (RR) was significantly higher in infected patients than in potentially non-infected participants (*p* = 0.001). Moreover, SpO_2_ was significantly lower among infected patients than in potentially non-infected participants (*p* = 0.03). Furthermore, 28 (53%) infected patients and 25 (47%) potentially non-infected participants took vitamin D supplements monthly. Additionally, three (6%) infected patients took zinc supplements, whereas no potentially non-infected participants did. However, there were no significant differences between the two study groups in terms of taking vitamin D and zinc supplements (data not shown in the table). 

### 3.2. Vitamin D Status and Sunlight Exposure of Infected Patients and Potentially Non-Infected Participants

The laboratory measurements were generally performed 7 ± 2 days after the RT-PCR test, and a statistically significant difference in days away was not found within the infected patients and potentially non-infected participants.

As represented in Table 2, we did not inspect the statistical significance in either 25(OH)D concentration or vitamin D status category between infected patients (26 ng/mL) compared with the potentially non-infected participants (29 ng/mL). More than a quarter of the potentially non-infected participants (i.e., 14 (26%) individuals) had vitamin D insufficiency (12–20 ng/mL); three (6%) individuals were deficient (<12 ng/mL), and 36 (68%) individuals had normal vitamin D (≤15 nmol/L).

The comparison of the 25(OH)D concentration between infected patients with *asymptomatic and mild illness* and patients with *moderate illness* is illustrated in Figure 2. We observed a marginally significant difference in terms of 25(OH)D concentration between patients with *moderate illness* (19 ± 12 ng/mL) compared to patients with *asymptomatic and mild illness* (29 ± 18 ng/mL) (*p* = 0.054).

Typical questions used to assess sunlight exposure are also listed in Table 2. The comparison of the components of sunlight exposure between infected patients and potentially non-infected participants revealed that the percentage of time spent in the shade was significantly higher in patients than in potentially non-infected participants. However, the other components did not show any significant differences between the two study groups. 

### 3.3. Zinc Status of the Infected Patients and Potentially Non-Infected Participants

As shown in Figure 3, infected patients showed a significantly lower serum concentration of zinc than potentially non-infected participants (101 ± 18 µg/dL in infected patients vs. 114 ± 13 µg/dL in potentially non-infected participants) (*p* = 0.013).

The comparison of the serum concentration of zinc between infected patients with *asymptomatic and mild illness* and patients with *moderate illness* is illustrated in Figure 4. The results demonstrated a lower serum concentration of zinc in patients with moderate illness (97 ± 17 µg/dL) compared to those with *asymptomatic and mild illness* (102 ± 18 µg/dL). However, this difference was not statistically significant (*p* = 0.412). 

### 3.4. Symptom Follow-Up, Outcomes, and Associations with Demographic, BMI and Laboratory Parameters

Table 3 shows the changes in clinical symptoms among infected patients from day 1 to 28 of the disease. The most common general, pulmonary, gastrointestinal, and neurologic symptoms at days 1 and 7 were fatigue, cough, anorexia, and smell disorder, respectively. Additionally, fatigue, cough, bloating, and smell disorder were the most common symptoms at days 14 and 21 of the disease. In addition, fatigue and sore throat, cough, constipation, and muscle pain were among the most common general, pulmonary, gastrointestinal, and neurologic symptoms on day 28 of the disease. Finally, all clinical symptoms except constipation and hypothermia showed a decreasing trend from days 1 to 28 of the disease.

Table 4 shows the results of the GEE model for the longitudinal relationship between vitamin D and zinc status at the second to seventh days of disease and clinical symptoms adjusted for age, sex, marital status, education levels, and BMI among infected patients with COVID-19.

The results revealed that the odds ratio of general symptoms of COVID-19 was three times higher among males than females (OR = 3.06; 95% CI, 1.13–8.33; *p* = 0.03). However, the odds ratio of neurologic symptoms in males was 0.41 times that of females (OR = 0.41; 95% CI, 0.17–0.98; *p* = 0.045). 

Furthermore, the patients who had normal vitamin D status were less likely to experience general symptoms and pulmonary symptoms than patients with vitamin D deficiency with ORs of 0.10 (95% CI, 0.04–0.24; *p* ≤ 0.001) and 0.27 (95% CI, 0.07–0.99; *p* = 0.05), respectively.

Additionally, the odds ratio of general symptoms of COVID-19 in patients with insufficient vitamin D was 0.2 times that of patients with vitamin D deficiency (OR = 0.19; 95% CI, 0.06–0.65; *p* = 0.008).

In the present study, all symptoms showed a decreasing trend over time. However, the marital status, education, age, BMI, and serum concentration of zinc variables were not significantly associated with clinical symptoms (*p* ≥ 0.05).

## 4. Discussion

This is likely the first study to characterize the association of vitamin D and zinc status with the severity and progression of symptoms of COVID-19. In this study, 53 outpatients with laboratory-confirmed COVID-19 disease and 53 potentially non-infected participants for whom the disease was excluded by RT-PCR were included to compare the vitamin D and zinc status in the body. In addition, the associations between the serum concentrations of 25(OH)D and zinc at the second to seventh days of disease and the progression of clinical symptoms among infected patients were evaluated by the GEE model adjusted for age, sex, marital status, education levels, and BMI. Our findings showed that in terms of vitamin D status, although the serum concentrations of 25(OH)D of infected patients and potentially non-infected individuals were statistically similar, a trend was noted for a lower serum concentration of 25(OH)D in moderate than asymptomatic or mild illness patients. 

One caveat to consider is that the patients with normal vitamin D status were less likely to experience general and pulmonary symptoms than patients with vitamin D deficiency. Additionally, patients with inadequate vitamin D status were less likely to report general symptoms of COVID-19 than patients with vitamin D deficiency. In other words, a normal vitamin D status at the second to seventh days of disease reduced the odds of general and pulmonary symptoms during the disease. Based on the results of the comparison between infected patients and potentially non-infected participants in terms of vitamin D status, it can be inferred that vitamin D status affects the severity of COVID-19 and the progression of symptoms during the clinical course of the disease.

Similarly, a recent study determined that the diagnosis of vitamin D deficiency could be useful in evaluating COVID-19 patients’ potential risk of disease development and severity [29]. A cross-sectional study in Qom City, Iran, in patients with COVID-19 reported a significant association between a hospital stay and a lower serum concentration of vitamin D. However, the correlation between vitamin D status and death rate (or the time interval to obtain a normal oxygen level) was not significant [30]. This may be due to the role of vitamin D in the immune system, from its receptors on the majority of immune cells to increase the production of anti-inflammatory cytokines versus pro-inflammatory cytokines or even the production of an antimicrobial peptide against enwrapped coronaviruses. Additionally, vitamin D upregulates angiotensin-converting enzyme 2 (ACE2) expression, which in the lungs has shown a protective effect against acute lung injury [31].

In addition, a case–control study confirmed that the serum concentration of vitamin D deficiency is associated with more severe lung involvement, longer disease duration, and the severity of radiologic pulmonary involvement as evaluated by computed tomography. In particular, serum concentration of 25(OH)D were significantly lower in COVID-19 patients with either multiple lung consolidations or diffuse/severe interstitial lung involvement than in those with mild involvement [32].

Moreover, a systematic review and meta-analysis concluded that a lower serum concentration of 25(OH)D accompanies severe presentation and mortality relating to COVID-19 disease [33]. A recent study described the relationship between vitamin D status and complications and mortality from COVID-19 in 46 countries. The results showed that the serum concentration of 25(OH)D in each country had a significant relationship with the number of deaths, the risk of being infected with SARS-CoV-2, and the severity of the disease [34]. Additionally, a short report in 20 European countries indicated that the serum concentration of 25(OH)D was also extremely low in elderly populations, especially in Spain, Italy, and Switzerland. It was also the most vulnerable population group in terms of COVID-19. This study concluded that vitamin D supplementation is recommended to protect against SARS-CoV-2 infection [12].

Although our results showed a trend for a lower serum concentration of 25(OH)D among moderate than asymptomatic or mild illness patients, these findings conflict with those of some previous studies showing strong protective effects of vitamin D. Moreover, a cross-sectional study conducted on biobank samples of participants from England, Scotland, and Wales showed that the serum concentration of 25(OH)D was associated with COVID-19 risk; however, this association disappeared after controlling for confounding factors [35]. These controversies suggest that further studies are needed to evaluate the protective effects of normal vitamin D status in COVID-19 patients.

Moreover, the evaluation of sunlight exposure and modifying factors among infected patients and potentially non-infected participants showed that the percentage of time spent in the shade was significantly greater in patients than in potentially non-infected participants. However, daily sun exposure, time spent outdoors, and time spent wearing a brimmed hat, long sleeves, and sunscreen did not indicate any significant differences between the two study groups. This result could support our finding that there is no significant difference between the two groups in terms of the serum concentration of vitamin D. 

The protective effect of the serum concentration of 25(OH)D on the severity of COVID-19 and the progression of symptoms during the clinical course of the disease might underlie some mechanisms: vitamin D has beneficial effects on protective immunity in part due to its effects on the innate immune system and β-cell function [7,9]. Immune cells express vitamin D receptors (VDRs). It is known that macrophages identify lipopolysaccharide (LPS), a surrogate for bacterial infection, through Toll-like receptors (TLRs). TLR binding increases the expression of both VDRs and 1-α-hydroxylase [36,37]. This results in the binding of the 1,25 D-VDR-RXR heterodimer to vitamin D response elements (VDREs), leading to the translocation of the complex into the cell nucleus, where it modifies the expression of hundreds of genes, including those involved in cytokine production [38]. The complex also induces the production of antimicrobial peptides, including cathelicidin and beta-defensin 4 [39]. These peptides co-localize within phagosomes with injected bacteria, as they disturb bacterial cell membranes and exhibit strong anti-microbacterial activity. The transcription of cathelicidin is dependent on sufficient 25(OH)D [37]. 

The administration of vitamin D in a dose of 5000 IU/kg has been shown to reduce the replication of rotavirus both in vitro and in vivo [40]. Vitamin D administration can also reduce the production of T helper type 1 (Th1) cell cytokines, such as interferon-γ and tumor necrosis factor-α (TNF-α), and the expression of pro-inflammatory cytokines by macrophages. It can also increase anti-inflammatory cytokine levels [41,42]. The induction of cytokine storms is also reduced by vitamin D. However, vitamin D supplementation did seem to non-significantly increase the risk of in-hospital mortality among COVID-19 patients addressing the maintenance of serum concentrations of vitamin D and zinc in a normal range to prevent the incidence or progression of clinical symptoms among COVID-19 patients [43]. 

The innate immune system generates both pro-inflammatory and anti-inflammatory cytokines in patients suffering from COVID-19 [1]. Binding to dipeptidyl peptidase-4 receptor (DPP-4/CD26) is one of the molecular virulence mechanisms employed by a coronavirus. It has been demonstrated recently that human DPP-4/CD26 interacts with the S1 domain of the SARS-CoV-2 spike glycoprotein [44]. In this context, vitamin D deficiency has been shown to remarkably reduce the expression of the DPP4/CD26 receptor in vivo [45]. Vitamin D is a strong inducer of autophagy [46] and inhibits HIV replication in macrophages via vitamin D-mediated induction of cathelicidin, perhaps by enhancing autophagy and phagosomal maturation [47]. 

In the present study, patients infected with SARS-CoV-2 had significantly lower serum concentrations of zinc than potentially non-infected individuals. However, the serum concentration of zinc was not different among COVID-19 patients with *mild or asymptomatic illnesses* compared to participants who had *moderate* COVID-19. Moreover, the serum concentration of zinc at the second to seventh days of disease showed no significant association with common clinical symptoms of COVID-19 in four categories during the period of day 1 to day 28 after the disease onset. 

Our finding is consistent with that of a prospective observational study conducted on COVID-19 inpatients at the time of hospitalization, which reported that the serum concentration of zinc was significantly lower in patients compared to healthy controls [48]. Additionally, a recent study in Turkey reported that in the first trimester of pregnancy, the serum concentration of zinc was significantly lower in pregnant women with COVID-19 compared to controls [49]. Moreover, a single-center study carried out on hospitalized patients with COVID-19 found that the serum concentration of zinc was significantly lower in patients who died than those who were admitted to an intensive care unit (ICU) or non-ICU and survived. However, contrary to our finding, the serum concentration of zinc at the time of admission could affect clinical outcomes in COVID-19 patients [50].

Additionally, as mentioned in a recently published review study, zinc may have beneficial effects including a decreased susceptibility to infection in the current and future pandemics [51]. In contrast to our study results, a review study revealed that a pre-existing severe zinc deficiency predisposes patients to a stronger progression of SARS-CoV-2 infections, and even a mild zinc deficiency should be corrected to prevent a more severe viral infection [5]. However, in the present study, no significant difference was observed between the serum concentration of zinc and the severity of COVID-19 disease, which may be due to insufficient sample size.

In terms of the effect of zinc during the COVID-19 pandemic, it is believed that zinc is a potential supportive treatment in therapy against COVID-19 disease due to its positive effects on the immune response [20]. 

Previous studies strongly revealed that zinc status is a critical factor that can influence antiviral immunity [52]. A meta-analysis of mostly high-quality studies by Aggarwal et al. [53] showed that the risk of lower respiratory tract infections or pneumonia and diarrhea or dysentery could be reduced in children after zinc administration. Additionally, a retrospective review reported that zinc supplementation at a total dosage of 2–2.5 mg/kg/day improved COVID-19 symptoms after 7 days of treatment. However, this study had some limitations, including the absence of blinding and a control group [54]. Moreover, an uncontrolled case series reported that the administration of a high dose of zinc salt oral lozenges for four consecutive outpatients with clinical characteristics of and/or laboratory-confirmed COVID-19 led to a significant improvement in symptomatic COVID-19 measures after one day of high-dose therapy, suggesting that zinc therapy played a role in clinical recovery [55].

It is thought that the supportive effects of zinc in patients with COVID-19 exist because of its immunomodulatory effects and several direct and indirect effects against a wide variety of viral species, predominantly RNA viruses [56,57]. It has been previously shown that the zinc cation (especially in combination with ionophore pyrithione) can inhibit the RNA polymerase of the SARS-CoV-2 virus, and this evidence makes zinc a potential therapeutic agent for patients with COVID-19 in combination with antiviral medications [57,58,59]. Accordingly, zinc can inhibit the elongation step of RNA transcription [57]. Zinc can induce its antiviral effects by suppressing RNA-dependent RNA polymerase (RDRP) and blocking the further replication of viral RNA as demonstrated for SARS-CoV-1 [60]. In addition, there is some evidence that suggests zinc can reduce ACE2 activity [31], which is the receptor for SARS-CoV-2 [61]. The modulation of antiviral immunity by zinc can also limit SARS-CoV-2 infection through the upregulation of interferon-alpha (IFN-α) production through the Janus kinase signal transducer and activator of transcription-1 (JAK/STAT1) signaling pathway in leukocytes [62] and increasing its antiviral activity [63]. In addition to its immunomodulatory effects, zinc, as an antiviral agent, exerts its beneficial roles and potential applications in the management of COVID-19, possibly by the enhancement of total antioxidant capacity [64].

Moreover, zinc has anti-inflammatory effects by blocking the inhibitor of nuclear factor kappa B (IκB) kinase (IKK) activity and subsequent nuclear factor kappa B (NF-κB) signaling, resulting in the downregulation of pro-inflammatory cytokine production [65,66]. On the other hand, a viral infection-related inflammatory response resulting in the overproduction of pro-inflammatory cytokines and cytokine storm is known to play a significant role in COVID-19 pathogenesis and patient outcomes [67]. Additionally, the coexistence of noncommunicable chronic diseases (NCDs) in COVID-19 patients may strengthen the inflammatory pathology and increase the risk for adverse outcomes and mortality [68]. In turn, inflammation can be under- or overestimated micronutrient deficiencies. Besides, zinc is a negative acute-phase reactant; therefore, inflammation accompanies serum hypozincemia [15,69]. Accordingly, the adjustment of zinc concentrations for inflammation is necessary when evaluating the zinc status among the population [69,70]. Several methods have been suggested to adjust for the effect of inflammation on the zinc status; however, to our knowledge, none have been examined in adults in whom chronic inflammation is common [70]. Additionally, there is no established agreement on how to control for the effect of inflammation on the serum concentration of zinc, which has a consequence for precise estimates of zinc status at the population level [69]. 

It is necessary to mention that our study covered a wide age range of participants, from children to the elderly population, who are among the high-risk groups for zinc deficiency. In addition, COVID-19 symptoms may exacerbate zinc deficiency, which is a threat to current high-risk groups [51]. Therefore, the cross-sectional nature of this study does not allow us to determine the causality relationship between zinc status and the progression of COVID-19 disease.

Adequate dietary intake of zinc and vitamin D could be considered as a possible solution to compensate for the low status of vitamin D and zinc, which to some extent may be effective on immunocompetence. However, vulnerable sections of populations may need supplements besides dietary advice to secure adequacy for these nutrients. In the case of low vitamin D status (<50 nmol/L), vitamin D supplementation (40 µg D3/day) is considered as an approach for the prevention of a destructive course of the inflammation induced by COVID-19. Moreover, a dietary zinc intake ≤ 25 mg/day was recommended as a preventive dose for COVID-19 on a long-term basis [21]. In addition, foods rich in zinc and zinc supplements could serve as adjuvants in combination with vaccines for the treatment of COVID-19 [64]. 

Our study has several limitations. First, only 53 laboratory-confirmed COVID-19 outpatients and 53 potentially non-infected participants were involved. Consequently, the small sample size has led to a cautious interpretation of the results. Second, as many of these findings are non-specific, they might overlap with other potentially coexisting deficiencies and illnesses. Our limited nutritional assessment suggests that other nutritional deficiencies might also affect the clinical signs and symptoms of COVID-19. Additional research in this area is needed. Third, recall bias is possible because data of clinical symptoms were self-reported. Fourth, the present study had a longitudinal component where the symptom progression of COVID-19 was observed, but there were no observations before the positive RT-PCR result.

However, the strengths of our study were the longitudinal nature and the follow-ups with the infected participants for one month, which helped us determine the relationship between the nutritional status of vitamin D and zinc at the second to seventh days of disease and the progression of clinical symptoms and recovery time. Additionally, we observed the differences between study groups after age and sex matching, as there is the belief that the difference between infected patients and potentially non-infected individuals might be affected by sex and age structure.

## 5. Conclusions

The results of the present study underline that although serum concentrations of 25(OH)D in infected patients and potentially non-infected participants were statistically similar, the role of vitamin D in the severity of COVID-19 was marginally significant. In addition, the severity of vitamin D deficiency is associated with the progression of general and pulmonary symptoms, indicating the importance of the evaluation of the vitamin D status at the onset of the disease as a relatively easy option to predict disease severity and the progression of COVID-19 symptoms. 

In terms of the zinc status, the results of the present study underline that patients with COVID-19 can have a lower serum concentration of zinc. However, the serum concentration of zinc was not different among COVID-19 patients with *mild or asymptomatic illness* when compared to participants who had *moderate* COVID-19. Moreover, serum concentrations of zinc at the second to seventh days of disease were not associated with the progression of symptoms among the COVID-19 patients. In other words, the serum concentration of zinc of the outpatients might not affect disease severity or the progression of symptoms.

Accordingly, serum concentrations of 25(OH)D and zinc should be examined in all inpatients and outpatients with COVID-19 and at different stages of the disease to maintain or promptly increase concentrations of 25(OH)D and zinc in the optimal range. Further studies are needed to confirm our findings.

## Figures and Tables

**Figure 1 nutrients-13-03368-f001:**
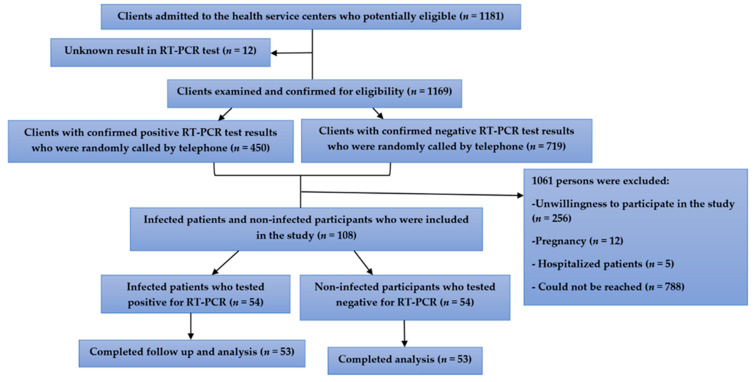
Flow diagram of the study design.

**Figure 2 nutrients-13-03368-f002:**
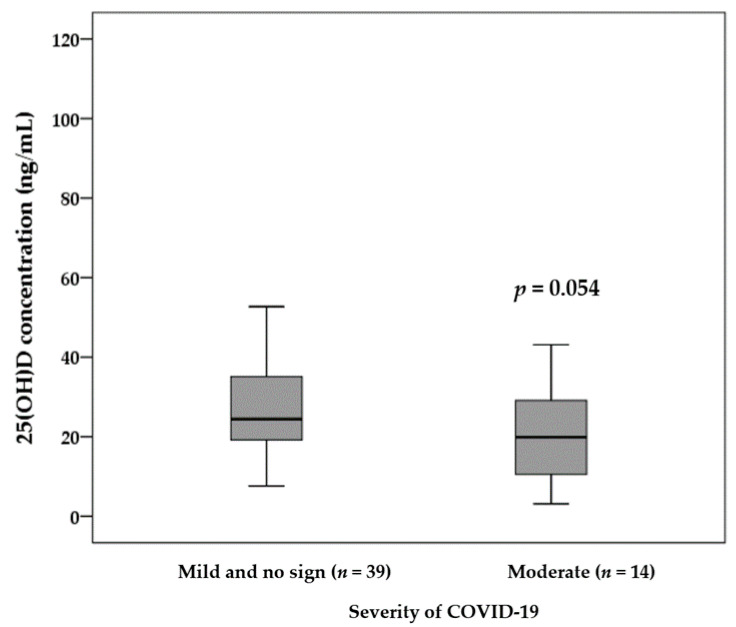
The comparison of the serum concentration of 25(OH)D between infected patients with different severity of COVID-19. An independent sample *t*-test was applied to analyze data. Patients with moderate COVID-19 showed a trend noted for a lower serum concentration of 25(OH)D than *mild and no sign illness* patients.

**Figure 3 nutrients-13-03368-f003:**
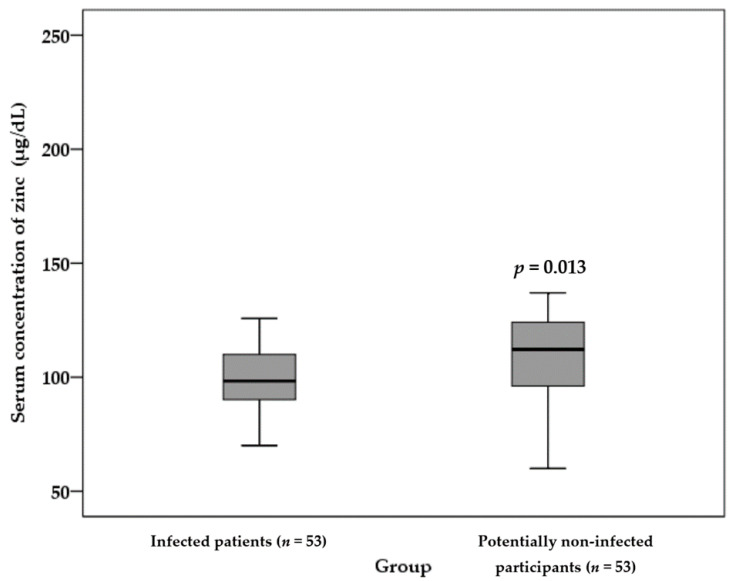
The comparison of the serum concentration of zinc between infected patients and potentially non-infected participants. An independent sample *t*-test was applied to analyze data. The serum concentration of zinc was significantly lower among infected patients than potentially non-infected participants.

**Figure 4 nutrients-13-03368-f004:**
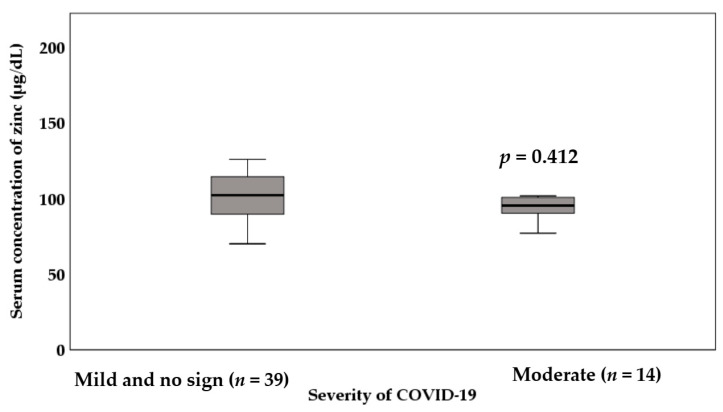
The comparison of the serum concentration of zinc between infected patients with different severity of COVID-19. An independent sample *t*-test was applied to analyze data. We did not find any significant difference between infected patients with *asymptomatic and mild illness* and patients with *moderate illness* in terms of serum concentration of zinc.

**Table 1 nutrients-13-03368-t001:** Demographic, clinical, comorbidity, and anthropometric characteristics of patients infected with SARS-CoV-2 and matched controls at the second to seventh days of disease ^1^.

Characteristics	Infected Patients (*n* = 53)	Potentially Non-Infected Participants (*n* = 53)	*p*-Value
Age (year)	41 ± 13	40 ± 14	0.609
Sex			
Male, *n (%)*	36 (68)	38 (72)	0.672
Married status, *n (%)*			
Single	12 (23)	12 (23)	0.592
Married	41 (77)	41 (77)	
Education levels, *n (%)*			
Illiterate	2 (4)	1 (2)	0.754
Under diploma	15 (29)	19 (37)	
Diploma	13 (25)	10 (19)	
College education	22 (42)	22 (42)	
Cigarette smoking, *n (%)*			
No	40 (76)	45 (85)	0.223
Yes	13 (25)	8 (15)	
RR (number/min)	14 ± 0.2	13 ± 0.3	0.001
PR (number/min)	91 ± 3	87 ± 2	0.271
SpO_2_ (%)	97 ± 1.4	97 ± 1.2	0.032
Duration of disease (day) ^2^	7 ± 2	-	
Comorbidities, *n (%)*			
Chronic pulmonary diseases	2 (4)	0 (0)	0.153
Hypertension	10 (19)	5 (9)	0.164
Diabetes mellitus	6 (11)	4 (8)	0.506
Obesity	13 (25)	21 (40)	0.096
Malnutrition	1 (2)	0 (0)	0.315
Cancer	2 (4)	0 (0)	0.153
Liver disease	5 (9)	3 (6)	0.462
Chronic neurological diseases	2 (4)	1 (2)	0.547
Chronic hematologic diseases	2 (4)	0 (0)	0.157
Renal diseases	3 (6)	4 (8)	0.696
Chronic heart disease	4 (8)	2 (4)	0.414
HIV	2 (4)	0 (0)	0.153
Asthma and allergy	6 (11)	5 (9)	0.750
Others ^3^	8 (15)	16 (30)	0.063
BMI (Kg/m^2^)	27 ± 5	28 ± 4	0.663

^1^ Independent sample *t*-test was conducted to analyze continuous variables, and the results were stated as mean ± standard deviation. Categorical variables were analyzed by chi-square test, and the results were presented as number (%). ^2^ Duration of disease indicates the number of days since the onset of the patient’s first clinical symptoms obtained by asking the infected patients and recording in the questionnaire. ^3^ Others including autoimmune disease, hemoglobinopathies, migraine, digestive system problems, hypothyroidism, hyperthyroidism, hyperlipidemia, endometrioses, neck, and back disk. BMI, body mass index; RR, respiratory rate; PR, pulse rate; SpO_2_, oxygen saturation.

**Table 2 nutrients-13-03368-t002:** Vitamin D status and characteristic items were used to measure individual sunlight exposure among patients infected with SARS-CoV-2 and potentially non-infected participants ^1^.

Components of Individual UV Exposure and Modifying Factors	Infected Patients (*n* = 53)	Potentially Non-Infected Participants (*n* = 53)	*p*-Value
25(OH)D (ng/mL)	26 ± 17	29 ± 16.	0.424
25(OH)D status, *n (%)*			
Vitamin D deficiency	10 (19)	3 (6)	0.086
Vitamin D insufficiency	9 (17)	14 (26)	
Normal vitamin D	34 (64)	36 (68)	
Daily sun exposure (minute)	78 ± 104	87 ± 60	0.585
How much time did you spend outdoors between the hours of 9 and 11 a.m.? (hour)	1.3 ±0.9	1.3 ±0.9	0.855
How much time did you spend outdoors between the hours of 11 a.m. and 1 p.m.? (hour)	1.2 ± 0.9	1.1 ± 1	0.810
How much time did you spend outdoors between the hours of 7 and 9 a.m.? (hour)	1.1 ± 1	1.1 ± 1	0.888
How much time did you spend outdoors between the hours of 1 and 3 p.m.? (hour)	0.8 ± 0.9	0.7 ± 0.9	0.594
How much time did you spend outdoors between the hours of 3 and 5 p.m.? (hour)	0.7 ± 0.9	0.6 ± 0.9	0.676
How much time did you spend outdoors between the hours of 5 and 7 p.m.? (hour)	0.8 ± 0.9	1 ± 0.9	0.328
What percent of this time did you spend under shade (e.g., tree or beach shade)? (%)	78 ± 22	63 ± 32	0.006
What percent of time did you wear a brimmed hat? (%)	18 ± 37	11 ± 31	0.330
What percent of time did you wear long sleeves? Long pants? (%)	84 ± 34	76 ± 39	0.264
What percent of time did you wear sunscreen? (%)	9 ± 26	2 ± 10	0.090

^1^ χ^2^ test for discrete and the independent sample *t*-test for continuous variables were applied to analyze data. The results have been shown with mean ± standard deviation for continuous and number (%) for discrete data. UV, ultraviolet.

**Table 3 nutrients-13-03368-t003:** Changes in clinical symptoms of COVID-19 among infected patients from day 1 to 28 of follow-up ^1^.

	Days of Follow-Up	Day 1	Day 7	Day 14	Day 21	Day 28
Clinical Symptoms						
General						
Fatigue		32 (60)	16 (30)	11 (21)	7 (13)	4 (8)
Fever		12 (23)	3 (6)	1 (2)	1 (2)	0(0)
Night sweats		25 (47)	10 (19)	8 (15)	2 (4)	0 (0)
Flushing		1 (2)	1 (2)	1 (2)	1 (2)	1 (2)
Chills		5 (9)	6 (11)	3 (6)	1 (2)	1 (2)
Hypothermia		0(0)	2 (4)	2 (4)	0 (0)	1 (2)
Runny nose		4 (8)	2 (4)	2 (4)	2 (4)	1 (2)
Sore throat		14 (26)	7 (13)	4 (8)	1 (2)	4 (8)
Pulmonary						
Chest pain		5 (9)	8 (15)	4 (8)	4 (8)	4 (8)
Shortness of breath		14 (26)	8 (15)	6 (11)	4 (8)	1 (2)
Cough		23 (43)	18 (34)	15 (28)	9 (17)	5 (10)
Gastrointestinal						
Anorexia		24 (45)	12 (23)	3 (6)	2 (4)	4 (8)
Abdominal cramps		10 (19)	9 (17)	4 (8)	3 (6)	1 (2)
Diarrhea		19 (36)	8 (15)	(0)	5 (9)	1 (2)
Vomiting		3 (6)	1 (2)	1 (2)	1 (2)	1 (2)
Nausea		11 (21)	3 (6)	2 (4)	1 (2)	1 (2)
Constipation		5 (9)	4 (8)	3 (6)	2 (4)	6 (11)
Bloating		8 (15)	7 (13)	6 (11)	3 (6)	1 (2)
Neurologic						
Headache		18 (34)	9 (17)	4 (8)	3 (6)	1 (2)
Muscle pain		11 (21)	7 (13)	6 (11)	3 (6)	5 (10)
Joint pain		13 (25)	4 (8)	4 (8)	2 (4)	1 (2)
Ear pain		5 (9)	5 (9)	3 (6)	1 (2)	2 (4)
Smell disorders		33 (62)	15 (28)	9 (17)	8 (15)	5 (9)
Taste disorder		26 (49)	11 (21)	5 (9)	2 (4)	3 (6)

^1^ Descriptive statistic were conducted to analyze data. Data were shown as *n (%).*

**Table 4 nutrients-13-03368-t004:** Estimates of observed symptom progression of COVID-19 and the association with demographic, BMI, and laboratory parameters among infected patients (*n* = 53) ^1^.

Symptom Categories				
Parameters	General	Pulmonary	Gastrointestinal	Neurologic
Age (year)	1.02 (0.99–1.05)	1.05 (1.00–1.11)	0.97 (0.92–1.02)	0.97 (0.93–1.02)
Sex				
Female	1 (ref)	1 (ref)	1 (ref)	1 (ref)
Male	3.06 (1.13–8.33) ^2^	1.11 (0.33–3.68)	1.71 (0.62–4.75)	0.41 (0.17–0.98) ^2^
Married status				
Single	1 (ref)	1 (ref)	1 (ref)	1 (ref)
Married	0.91 (0.23–3.54)	1.32 (0.39–4.42)	2.56 (0.87–7.32)	1.97 (0.60–6.69)
Education levels				
Illiterate	1 (ref)	1 (ref)	1 (ref)	1 (ref)
Under diploma	0.70 (0.08–5.96)	18.45 (0.73–463.77)	0.28 (0.03–3.08)	0.10 (0.01–1.38)
Diploma	0.54 (0.06–5.10)	7.45 (0.21–212.65)	0.29 (0.02–4.12)	0.26 (0.02–3.35)
College education	1.24 (0.17–9.09)	12.80 (0.46–354.25)	0.64 (0.06–7.05)	0.38 (0.03–4.69)
Category of vitamin D status				
Deficiency	1 (ref)	1 (ref)	1 (ref)	1 (ref)
Insufficiency	0.19 (0.06–0.65) ^4^	0.63 (0.10–3.87)	0.52 (0.10–2.81)	1.09 (0.32–3.72)
Normal	0.10 (0.04–0.24) ^3^	0.27 (0.07–0.99) ^2^	0.39 (0.12–1.21)	0.50 (0.21–1.19)
Serum concentration of zinc (µg/dL)	0.98 (0.96–1.01)	1.00 (0.98–1.02)	0.99 (0.97–1.01)	1.02 (0.99–1.05)
BMI (Kg/m^2^)	1.06 (0.97–1.17)	0.98 (0.90–1.07)	1.04 (0.95–1.15)	1.08 (0.97–1.19)
Time (Day)	0.88 (0.84–0.93) ^4^	0.91 (0.88–0.95) ^4^	0.91 (0.87–0.94) ^4^	0.89 (0.86–0.93) ^4^

^1^ Odds of common clinical signs and symptoms of COVID-19 followed in days 1 to 28 of disease (95% CI). General estimation equation (GEE) was applied to analyze data. ^2^
*p* < 0.05, ^3^ *p* < 0.01, ^4^ *p* < 0.001.

## Data Availability

The data presented in this study are available on request from the corresponding author.

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
