# Peer review of "The Association between Vitamin D and Zinc Status and the Progression of Clinical Symptoms among Outpatients Infected with SARS-CoV-2 and Potentially Non-Infected Participants: A Cross-Sectional Study"

_nutrients, 2021, doi:10.3390/nu13103368_

Round 1

Reviewer 1 Report

Dear editor(s) and authors,

First of all, I would like to thank the editor of this high-impact journal for inviting me to review this interesting article entitled “The Association between Vitamin D and Zinc Status with the Disease Onset and Progression of Clinical Symptoms among Outpatients Infected with COVID-19 and Non-Infected Participants: A Cross-Sectional Study”.

This is a well-written paper by Golabi et al. that an association between the status of Vitamin D and Zinc and the progression of COVID-19 symptoms in a cohort of 53 cases (infected) and 53 controls (non-infected) are described. Therefore, this is a study of high interest given the importance of the pandemic and is adapted to the topic of the journal, since it deals with nutritional-health issues. Despite only analyzing data from 106 subjects, the approach is well focused on clarifying the hypothesis raised.

However, some aspects of the article generate doubts and there are minor details that must be corrected.

My comments separated by article sections are listed below:

Abstract

  • Writing sections within the abstract are not mandatory (background, methods,) and can help authors save words. I recommend removing them.
  • The authors talk about "uninfected" subjects (control group), but are based on the result of a specific PCR test, so it is unknown if (any of) these individuals have unknowingly suffered from the disease. Therefore, I recommend that the authors be more cautious and always accompany the qualification of "not infected" with the word "potentially" or "PCR negative". This applies to the entire article.
  • In line 27, clarify that the levels of Vitamin D are not lower, but that there is a trend. It should be noted.
  • The term odds ratio can be defined as OR, and then use that abbreviation.

Introduction

  • The summary is well broken down and the context and relevance of the two micronutrients is explained. However, the authors only focus on the body status of these micronutrients and neglect the importance of acquiring them through diet. There is already proven evidence that low levels (without reaching deficiency) of Vitamin D and Zinc intake (from food) are negatively correlated with incidence and mortality from COVID-19. Here are some references that the authors should comment on and cite, in my opinion (DOI: 10.3390/nu12092738; DOI: 10.3390/nu12082358; DOI: 10.1016/j.ijid.2020.08.018).
  • The objective of the study to be pursued must be clearer (better specify), and not only point out that associations between parameters will be analysed.

Materials and methods

Materials and methods are well described and help a lot to understand the results. However, there are some comments:

  • The role of the subjects considered "controls" may contain artifacts, since it is not known if they are convalescing from the disease with negative PCR. Can the authors provide data on serological tests to confirm the suitability of these volunteers as real controls?
  • On line 187, the authors should better specify how the cut-off points were established, what are the criteria and why these thresholds? This information should be available without having to resort to appointment 22. On the other hand, what happens with the upper thresholds and with the values that exceed this threshold? How are these data processed?
  • In section 2.7, what happens with the data that do not comply with normality? Have they been transformed or are they no longer used? What about the homogeneity of variances? In student's T test, is Levene applied?

Results

The results are well described. I have some suggestions for in this section:

  • Please, the decimals in the tables should be standardized
  • The authors separate infected patients with asymptomatic and mild illness from those with moderate illness. What happens if asymptomatic and mild illness are analyzed separately?
  • Authors must specify in the graphs the N in each group.
  • The statistical assessment carried out with the Student's T seems to me not very robust. I recommend that the authors carry out another type of statistic adjusting for the covariates specified in section 2.7 to "validate" the results obtained in the graphs of the results. One option may be regression analysis.

Discussion

The discussion is good, where the authors try to explain the biological plausibility of their results through the support of the literature. Here, as I have done for the introduction, I recommend that the authors focus on possible solutions to compensate for the low status of these two micronutrients: show that the bibliography supports that the low intake of Vitamin D and Zinc is also associated with a higher incidence and severity of the COVID-19 (in the introduction I have written some potential cites for the authors that support this fact).

Reviewer 2 Report

Abstract

Line 25-26- This sentence is unclear; do you mean an association of clinical symptoms with vitamin D and zinc status?

Line 31-32- admission? Better mention which day of infection

Introduction

Material and Methods

The authors need to make several things clear in the method section.

How did the author identify the stage of COVID infection? Isn’t it possible the baseline clinical symptoms recorded were on patients in different stage of COVID infection? It is very likely patients were in early, mid or late stage of COVID disease. How did the author rule this out? Was SARS-CoV-2 antibodies level measured in COVID patients? Are there chances that the non-infected controls might have recovered COVID recently? How to rule this out?

During the time frame June 6,2020 to August 12, 2020, how many COVID patients visited the center?

The clinical symptoms were recorded at the time of COVID test (called day 1), day 7, 14, 21 and 28. The blood samples used to estimate vitamin D and zinc status were done at around day 7. In this case, the micronutrient status would give day status at day 7 of COVID disease. Has the author collected data on patients’ dietary habit that would suggest that there were no changes in dietary pattern in these 7 days?

Line 180-181- Grammar issue

Line 194-195- Gramma issue in this sentence.

Results

Table 1: Does the baseline here indicates day 1 when subjects underwent screening at COVID center? If so, how is the duration of infection 7 days? Please correct.

Was RR, PR and Spo2 data also collected repeatedly until 28 days? Please mention in method section.

The baseline vitamin D and zinc were measured at around 7th day after first identification of COVID infection. How likely are these values to deviate from what it would be at time of first COVID infection identification? Would it effect on the significance of vitamin D status between infected versus non-infected? The authors need to correct this throughout the manuscript that blood vitamin D and zinc status were day 7 status (while COVID was progressing) not the baseline when COVID started.

Figure 2- I wouldn’t put asterisk for p of 0.054.

Figure 4- why asterisk when there is no significant difference?

On what basis the authors classified symptoms as general, pulmonary or neurological?

The authors need to show changes in symptoms in COVID patient longitudinally from day 1 to 28 in a table in result section. Which were the most common symptoms at day 1, 7, 14, 21 and 28? Please mention.

Discussion

Line 342-343: I wouldn’t call it baseline vitamin D and Zinc

Line 352- If the usual cycle of COVID is 14 days, then day 7 would not be the beginning of disease.

Reviewer 3 Report

REVIEW OF:
The Association between Vitamin D and Zinc Status with the Disease Onset and Progression of Clinical Symptoms among Outpatients Infected with COVID-19 and Non-Infected Participants: A Cross-Sectional Study
Sahar Golabi 1 , Maryam Adelipour 2 , Sara Mobarak 3 , Maghsud Piri 4 , Maryam Seyedtabib 5 , Reza Bagheri 6 , Katsu- 6 hiko Suzuki 7 , Damoon Ashtari Larki 8 , Fatemeh Maghsoudi 9 and Mahshid Naghashpour 10 *

COMMENTS/QUESTIONS

1. Abstract: “This study revealed the importance of 25(OH)D measurement as a relatively easy option to predict the progression of general and pulmonary symptoms.“ COMMENT: on clinical ground, why bothering to order vit D and zinc tests (indirect makers of the severity of infection) that are not included in the prognostic guidelines, whereas ordering reliable blood inflammatory tests along with the imaging and coagulation makers are indeed established clinical prognosticators in COVID-19. QUESTION: what the authors mean by “general” symptoms?
2. This study established association between vitamin D/zinc deficiency and severity of COVID-19, which is NOT a new knowledge.
3. What was the Ct value of PRC tests and where did the reagents and primers come from?
4. There is no info about the measurement of viral load in the study cohort.
5. No inflammatory markers for comparison purpose were included in this study.
6. No lung imaging for the assessment of the severity of viral illness and the correlate of vit D/zinc deficiency was included.
7. It is difficult to evaluate the aim of this paper without comparable blood markers other than Vit D and zinc, or even without imaging impressions.
8. It appears this study only focused on mild to moderate patients, lack of data on severe or critical cases.
9. Table 1. What is the significance of underdiploma (uneducated) vs. diploma (educated)? For example, if an educated person spreads the virus to a community of uneducated persons, does this mean the uneducated community is responsible for community transmission? In a pandemic circumstance, all sectors of the society are affected unless certain sector of the society don’t have access to prevention programs.
10. This study failed to establish a clinical definition of non-infected study participants (placebo group).

11. Discussion:
a. “This is likely the first study to simultaneously characterize the association of vitamin D and zinc status with onset, severity, and progression of symptoms of COVID-19.” COMMENT: What if a person has vit D deficiency, but not zinc deficiency, or vise vera? The word “simultaneously” seems ambiguous in view of the type of data presented in this study.
b. “Our findings showed that in terms of vitamin D status, although serum concentration of vitamin D of infected patients and non-infected individuals were statistically similar, the role of vitamin D status in the severity of COVID-19 symptoms was marginally significant.” COMMENT: This statement is in contrast with the title of this paper (e.g., Progression of Clinical Symptoms among Outpatients Infected with COVID-19 – see the title).
c. “A cross-sectional study in Qom City, Iran, in patients with COVID-19 reported a significant association between hospital stay and lower serum concentration of vitamin D.” COMMENT: What factor(s) during hospital stay can cause lower serum concentrations of vitamin D?
d. “However, although our results showed ……………” COMMENT: what kind of English is this, using both however and although?
e. “Some mechanisms might underlie the protective effect of serum concentration of 25(OH)D on the severity of COVID-19 and progression of symptoms during the clinical course of the disease.” COMMENT: For example, what mechanism?
f. “However, the strengths of our study were the longitudinal nature and the follow-ups with the infected participants for one month, which helped us determine the relationship between the nutritional status of vitamin D and zinc at the onset of the disease and the progression of clinical symptoms and recovery time.” COMMENT: to measure an accurate level of vit D, the subject must not take supplemental vit D one week prior to the test. Was this protocol followed?

12. Estimation of serum vit D and zinc levels may be a prudent approach in COVID-19 patient prognosis, but such supplements have no significant established therapeutic role during acute viral infection unless the authors provide a peer-review reference. Meta-analyses of RCTs reveal “protective” effects of vitamin D against acute respiratory infections, albeit these effects were of modest size and with substantial heterogeneity. Further, why bothering to measure serum vit D and zinc levels in a mild infection? What would be the clinical rationale? 13. With regards to autophagy indicated in the discission section, vit. D “metabolites” have long been known to support innate antiviral effector mechanisms, including induction of antimicrobial peptides and autophagy.
14. What % of Iranian population are vit. D and zinc deficient and where in the country such deficiencies are most common?
15. There is no compelling data to show an association between vit D and zinc status with the “disease onset.” What clinical criterion did the authors used to establish disease onset? Considerations should be taken that genetic factors also play a role in bioavailability and status of vit. D, and zinc.
16. What genetic variant of SARS-CoV-2 was tested (RT-PCR) in this study?

17. Conclusion: “Accordingly, serum concentrations of 25(OH)D and zinc should be examined in all inpatients and outpatients with COVID-19 and at different stages of the disease to maintain or promptly increase concentrations of 25(OH)D and zinc in the optimal range.” COMMENT: In the western countries, it takes about 48 hours to obtain the test results of 25(OH)D for outpatient setting. What if during this period, the patient progresses to moderate or severe disease?

Round 2

Reviewer 1 Report

The authors have answered and modified propoerly my suggestions accordingly. The problem of "controls" remains unsolved, as the authors claim that there are no "valid" serological tests in their country. I understand that it is an insurmountable point... however in my opinion the results obtained in this article deserve to be published.

Reviewer 2 Report

I am happy with the answers in response to my queries and subsequent corrections made in the revised version by the authors.

Author Response

This manuscript is a resubmission of an earlier submission. The following is a list of the peer review reports and author responses from that submission.

Round 1

Reviewer 1 Report

This study recruited 53 COVID-19 outpatients and 53 sex- and age-matched healthy subjects. The aim was to investigate whether baseline vitamin D and zinc status serum levels could predict disease severity. The data showed that the patients with lower vitamin D serum levels at baseline later developed the more severe disease (i.e., moderate illness). However, there was no difference in baseline serum vitamin D levels between infected and non-infected subjects. In addition, serum zinc levels were significantly lower in infected patients compared to non-infected subjects. But, the patients with lower zinc serum levels at baseline later did not develop the more severe disease. The study is interesting. However, there are several significant concerns:

  1. Although stating that “infected patients were categorized according to disease severity and prognosis using CDC criteria,” the author mentioned that “the clinical symptoms of COVID-19 among infected patients were estimated as self-reported from the infected patients.” It appeared that no follow-up hospital examinations, such as clinical assessment, imaging, and SpO2 were performed.  The accuracy of pure self-reporting symptoms was a significant concern.
  2. No infected patients with severe or critical diseases were recruited.
  3. The sample size was tiny.
  4. It is now known why only 108 subjects were recruited from 1169 eligible individuals.
  5. As has been cited by the authors, other reports have shown that lower vitamin D status may predict COVID-19 severity.

Reviewer 2 Report

Authors submitted a well written and well analyzed manuscript.  The study limitations were correctly identified.  One potential bias that may occur is reverse causation, e.g. where COVID-19 causes the nutritional deficiency rather than the deficiency occurring before appearance of COVID-19.  Study had a longitudinal component where disease severity could be observed, but there were no observations prior to positive PCR result.  Authors need to bring this up as a potential limitation to the study.

Reviewer 3 Report

In this manuscript, the authors report the association between vitamin D, zinc status and severity of COVID19. I find the manuscript well discussed and very interesting for the community, adds additional information for the importance of vitamin D and zinc regarding COVID19, but there are some matters that have to be addressed.

As the authors described in the discussion: ”This is likely the first study to simultaneously characterize the association of vitamin D and zinc status with onset, severity, and progression of symptoms of COVID-19" . If I understand it rightly, the blood samples were collected after the PCR-test (6.5±1.9 days after the RT-PCR test), for me it is not clear how the association of vitamin D or zinc status with onset of COVID19 were conducted. Is there a time-dependent factor in the collection?  Are zinc or vitamin D levels declining over time with increase in severity?

The Inclusion and exclusion criteria is for me not clear. As it was described in line 123-125: “Pregnant and lactating 123 women, subjects with uncertain RT-PCR test results, cigarette smokers, and patients with 124 sickle cell anemia or thalassemia were excluded”, but it is shown in Table 1 that, 24,5% infected patients and 15.1% controls are smoker.

Furthermore, almost half of patients and control have the vitamin D supplementation, maybe with different dose and duration. That makes the association analysis of vitamin D in COVID-19 not convinced enough for me. Also, as described in line 141: “In the present study, no infected patients with severe or critical diseases were found among the participants.” It makes  difficult in this study to discuss the association of the nutrients status with severity of COVID-19.

Round 2

Reviewer 1 Report

  1. The authors cited a CDC report to validate the felt-report design. Although self-report data may provide some insights, the data may not provide sufficient details on biological processes.
  2. The authors provided details of their power analysis. The sample size may apply for the two general categories, i.e., infected patients and non-infected participants. However, the manuscript is based on two main comparisons. One comparison is between the patients with moderate illness and those patients with asymptomatic or mild disease. The other comparison is between patients with normal and insufficient vitamin D status and those with vitamin D deficiency. They only have three vitamin D deficient patients (Table 2). Therefore, the study is severely underpowered.

Author Response

Manuscript ID nutrients-1314683

Response to Reviewer 1 (Round 2)

Dear editor in chief of the Nutrients journal,

Thank you for giving us the opportunity to submit the second revision of the manuscript “A Case-Control Study of Vitamin D and Zinc Status in Outpatients Infected with COVID-19 and Non-Infected Participants: Association with the Progression of Symptoms During the Clinical Course of the Disease” for publication in the Nutrients Journal. We appreciate the time and effort that you and the reviewers devoted to providing feedback on our revised manuscript and are thankful for the perceptive comments on and valuable improvements to our manuscript. We have included most of the suggestions made by reviewer 1. Those changes are highlighted within the manuscript using blue-colored text for the corrections according to the comments of reviewer 1. Please see below, in red, for a point-by-point response to reviewer 1's comments and concerns.

We tried to ensure that our response letter adequately addressed each of reviewer 1's comments.

Sincerely,

Dr. Mahshid Naghashpour, Ph.D. (Nutrition)

Response to Reviewer 1 Comments

Comments and Suggestions for Authors

Point 1: The authors cited a CDC report to validate the felt-report design. Although self-report data may provide some insights, the data may not provide sufficient details on biological processes.

Response 1: Thank you for pointing this out. Although we agree that this is an important
consideration, however, we should explain that we got ideas from studies conducted during the Corona pandemic to determine a practicable, feasible, and safe procedure for researchers to have minimal exposure to patients with COVID-19. In studies conducted during the Corona Pandemic, published in different journals, aiming to assess disease progression has been also applied self-report methods to collect the patients' health information including symptoms, hospitalization, reverse-transcription PCR (RT-PCR) test outcomes, demographic information, and pre-existing medical conditions (Menni et al. study), current mental health (Taylor et al. study), and COVID-19 test status, COVID-19- associated symptoms, and self-perceived health status (Peters et al. study).

 Menni, C., Valdes, A.M., Freidin, M.B.,  Sudre, C.H.,  Nguyen, L.H., Drew, D.A., et al. Real-time tracking of self-reported symptoms to predict potential COVID-19. Nat Med 2020, 26, 1037–1040. DOI: 10.1038/s41591-020-0916-2

Taylor,  S., Landry, C. A., Paluszek, M. M., Fergus, T. A., McKay, Asmundson, D. G.J. G. COVID stress syndrome: Concept, structure, and correlates. Depress Anxiety 2020, 37, 706-714. DOI: 10.1002/da.23071. 

Peters, A.,  Rospleszcz, S.,  Greiser, K. H., Dallavalle, M., Berger, K. The Impact of the COVID-19 Pandemic on Self-Reported Health Early Evidence from the German National Cohort. Dtsch Arztebl Int 2020, 117, 861–867. DOI: 10.3238/arztebl.2020.0861

 Besides, we rewrote the description of self-reported data collection with more details in the “Data sources and measurements” subsection on page 4, paragraph 8, lines 1-5

Point 2: The authors provided details of their power analysis. The sample size may apply for the two general categories, i.e., infected patients and non-infected participants. However, the manuscript is based on two main comparisons. One comparison is between the patients with moderate illness and those patients with asymptomatic or mild disease. The other comparison is between patients with normal and insufficient vitamin D status and those with vitamin D deficiency. They only have three vitamin D deficient patients (Table 2). Therefore, the study is severely underpowered.

 Response 2:  While we appreciate the reviewer’s feedback, we respectfully disagree. The purpose of the comparison between patients with normal and inadequate vitamin D status and those patients with vitamin D deficiency was to determine the association between the severity of vitamin D deficiency with the onset of the COVID-19 disease. As seen in the SPSS software output, 0 cells (0.0%) have an expected count of less than 5 and the minimum expected count is 6.50. This shows that 3 individuals with vitamin D deficiency in the non-infected participants' group do not cause any problems and were underpowered in the analysis, statistically. On the other hand, in the infected patients' group, 10 individuals had vitamin D deficiency.

However, as stated in the “Discussion” section, the small sample size has led to a cautious interpretation of the results.

Crosstabs

Case Processing Summary

Cases

Valid

Missing

Total

N

Percent

N

Percent

N

Percent

Vitamin D category * Group

106

66.3%

54

33.8%

160

100.0%

Vitamin D category * Group Crosstabulation

Group

Total

Infected patients

(n=53)

Non-infected participants (n=53)

Vitamin D category

vitamin D deficiency

Count

10

3

13

Expected Count

6.5

6.5

13.0

% within Vitamin D category

76.9%

23.1%

100.0%

Vitamin D insuficiency

Count

9

14

23

Expected Count

11.5

11.5

23.0

% within Vitamin D category

39.1%

60.9%

100.0%

normal vitamin d

Count

34

36

70

Expected Count

35.0

35.0

70.0

% within Vitamin D category

48.6%

51.4%

100.0%

Total

Count

53

53

106

Expected Count

53.0

53.0

106.0

% within Vitamin D category

50.0%

50.0%

100.0%

Chi-Square Tests

Value

df

Asymptotic Significance (2-sided)

Pearson Chi-Square

4.913a

2

.086

Likelihood Ratio

5.129

2

.077

Linear-by-Linear Association

1.533

1

.216

N of Valid Cases

106

a. 0 cells (0.0%) have an expected count of less than 5. The minimum expected count is 6.50.
